# Prospective association between adherence to the Mediterranean diet and hepatic steatosis: the Swiss CoLaus cohort study

Saman Khalatbari-Soltani [1,2,3] Pedro Marques-Vidal [3] Fumiaki Imamura [4] Nita G. Forouhi [4]

► Prepublication history and additional materials for this paper are available online. To view these files, please visit the journal online (http://dx.doi.org/10.1136/bmjopen-2020-040959).

FI and NGF contributed equally.

FI and NGF are joint senior authors.

For numbered affiliations see end of article.

**Correspondence to**
Dr Saman Khalatbari-Soltani; saman.khalatbarisoltani@sydney.edu.au and Professor Nita G. Forouhi; nita.forouhi@mrc-epid.cam.ac.uk

## ABSTRACT

**Objective** The Mediterranean diet has been promoted as a healthy dietary pattern, but whether the Mediterranean diet may help to prevent hepatic steatosis is not clear. This study aimed to evaluate the prospective association between adherence to the Mediterranean diet and risk of hepatic steatosis.

**Design** Population-based prospective cohort study.

**Setting** The Swiss CoLaus Study.

**Participants** We evaluated 2288 adults (65.4% women, aged 55.8±10.0 years) without hepatic steatosis at first follow-up in 2009–2012. Adherence to the Mediterranean diet was scaled as the Mediterranean diet score (MDS) based on the Mediterranean diet pyramid ascertained with responses to Food Frequency Questionnaires.

**Outcome measures** New onset of hepatic steatosis was ascertained by two indices separately: the Fatty Liver Index (FLI, ≥60 points) and the non-alcoholic fatty liver disease (NAFLD) score (≥−0.640 points). Prospective associations between adherence to the Mediterranean diet and risk of hepatic steatosis were quantified using Poisson regression.

**Results** During a mean 5.3 years of follow-up, hepatic steatosis was ascertained in 153 (6.7%) participants by FLI criteria and in 208 (9.1%) by NAFLD score. After multivariable adjustment, higher adherence to MDS was associated with lower risk of hepatic steatosis based on FLI: risk ratio 0.84 (95% CI 0.73 to 0.96) per 1 SD of MDS; 0.85 (0.73 to 0.99) adjusted for BMI; and 0.85 (0.71 to 1.02) adjusted for both BMI and waist circumference. When using NAFLD score, no significant association was found between MDS and risk of hepatic steatosis (0.95 (0.83 to 1.09)).

**Conclusion** A potential role of the Mediterranean diet in the prevention of hepatic steatosis is suggested by the inverse association observed between adherence to the Mediterranean diet and incidence of hepatic steatosis based on the FLI. The inconsistency of this association when hepatic steatosis was assessed by NAFLD score points to the need for accurate population-level assessment of fatty liver and its physiological markers.

## INTRODUCTION

Hepatic steatosis is the most common cause of liver disease.[1] In westernised countries, hepatic steatosis affects up to 34% of the

### Strengths and limitations of this study

► This study had the benefit of a relatively large sample size and an average of 5.3 years of follow-up.

► We applied a definition of the Mediterranean diet that has been shown to be valid in a non-Mediterranean population.

► Our ascertainment of hepatic steatosis was based on two indices that have been validated for use in large epidemiological studies.

► We used dietary data measured only once at baseline, and intraindividual variation over time might be present which may weaken the observed associations towards the null; however, dietary intake in CoLaus was relatively stable, suggesting that lack of repeated dietary measures is unlikely to alter our findings substantially.

► The precision of the two hepatic steatosis indices used is different and may be influenced by the presence of steatohepatitis or advanced liver fibrosis.

general population and up to 74% of obese individuals, depending on the definition used.[2–4] Hepatic steatosis—fat content of more than 5% of liver volume—is the first recognisable stage of non-alcoholic fatty liver disease (NAFLD).[1] Hepatic steatosis, particularly NAFLD, may progress to end-stage liver disease including fibrosis, cirrhosis and hepatocellular carcinoma.[5] Moreover, as hepatic steatosis increases the risk of metabolic syndrome, type 2 diabetes and cardiovascular disease (CVD), its prevention is of public health importance.[6] An unhealthy dietary pattern remains one of the primary targets of lifestyle modification for the prevention and management of hepatic steatosis and NAFLD.[7 8]

The Mediterranean diet has been recently recommended for treatment of NAFLD.[9] In recent years, a growing body of evidence supports the idea that the Mediterranean diet may be the reference nutritional profile

for the prevention of hepatic steatosis development.[10–12] Adherence to the Mediterranean diet has been reported to have a beneficial impact on risks of CVD,[13 14] type 2 diabetes[15] and metabolic syndrome.[16] Trial evidence demonstrated the potential benefits of the Mediterranean diet against progress of hepatic steatosis focusing on individuals with existing hepatic steatosis, either alone[17–22] or associated with metabolic risk factors such as obesity or diabetes.[22–25] Research among those without clinically manifest hepatic steatosis is restricted to observational evidence, reporting an inverse association that greater adherence to a Mediterranean diet is associated with lower prevalence of hepatic steatosis.[26 27] However, the cross-sectional design of these studies limits inference for causal associations and can be used mainly for hypothesis generation. Relevant longitudinal evidence for the primary prevention of hepatic steatosis or NAFLD has been reported only by the Framingham Heart Study, with a significant inverse association of adherence to the Mediterranean diet with risk of hepatic steatosis in 1521 adults over 6 years of follow-up,[28] but evidence is lacking in Europe.

Given the limited evidence from population-based epidemiological studies thus far, we aimed to investigate the prospective association between adherence to the Mediterranean diet and the risk of developing hepatic steatosis among adults without clinically manifest hepatic steatosis. We hypothesised that greater adherence to the Mediterranean diet would reduce the risk of hepatic steatosis.

## METHODS

### Study population

We evaluated participants in the CoLaus Study, an ongoing population-based cohort investigating the clinical, biological and genetic determinants of CVD in the city of Lausanne, Switzerland.[29] Inclusion criteria of the recruitment were adults of European origin, aged 35–75 years.[29] There were three study phases: baseline recruitment in 2003–2006 (n=6733), the first follow-up in 2009–2012 (n=5064) and the second follow-up in 2014–2017 (n=4881). We conducted dietary assessment at the first follow-up and therefore considered the first follow-up as the study baseline. Fatty Liver Index (FLI) and NAFLD score, two indices of hepatic steatosis, were available at baseline.[27] If participants met the joint criterion of FLI ≥60 or NAFLD score ≥−0.640 at baseline, we excluded them as prevalent cases (see below) (n=2036). We also excluded participants with missing information on diet, outcome and covariates (n=740) (online supplemental figure S1).

### Patient and public involvement

Patients and/or the public were not involved in the design, conduct, reporting or dissemination plans of this research.

### Dietary assessment

Participants completed a self-administered, 97-item, semiquantitative Food Frequency Questionnaire (FFQ) about their habitual dietary intake over the last 4 weeks,[30] the validity of which had been assessed in canton Geneva against 24-hour recalls.[30 31] For each item, participants were instructed to report consumption frequencies by selecting one of the seven frequency options from 'less than once during the last 4 weeks' to 'two or more times per day' and by selecting a usual serving size (smaller, equal or bigger to a reference size).

### Mediterranean diet scores

We derived the pyramid-based Mediterranean diet score (MDS) as a measure of adherence to the Mediterranean diet from responses to the FFQ as we conducted previously.[27] This MDS is based on the Mediterranean dietary pyramid proposed by the Mediterranean Diet Foundation for both Mediterranean and non-Mediterranean countries, and accounting for the traditional Mediterranean diet, contemporary lifestyle and food environment.[32] We have previously reported that this MDS scoring algorithm predicted CVD incidence,[33] as well as the prevalence of hepatic steatosis[27] in non-Mediterranean populations. Briefly, a continuous score of 0–1 was assigned for each recommended level of the 15 components of the pyramid (vegetables, legumes and fish as healthy items; red meat, processed meat, potato and sweets as unhealthy items; and fruits, nuts, cereals, eggs, dairy, white meat and alcoholic beverages as items for which moderate consumption was recommended). The resulting MDS ranges between 0 and 15 on a continuous scale. The MDS calculation was adjusted to an energy intake of 2000 kcal/d (8.37 MJ/day) by applying a regression-residual technique for energy adjustment to each food group variable.[33 34]

### Ascertainment of hepatic steatosis

Two indices of hepatic steatosis were evaluated: FLI[35] and NAFLD liver fat score.[36] FLI was calculated based on a logistic function including body mass index (BMI), waist circumference, fasting triglycerides and gamma-glutamyl transferase (GGT) levels as follows:

$$FLI = 1/(1+e^{-(0.953 \times \ln(\text{triglycerides}) + 0.139 \times BMI + 0.718 \times \ln(\text{GGT}) + 0.053 \times \text{waist circumference} - 15.745)}).$$

FLI×100 ranges from 0 to 100. Presence of hepatic steatosis was defined by FLI ≥60, a value with a sensitivity of 61% and a specificity of 86%.[35] FLI was tested previously in comparison to ultrasonography with an area under the receiver operating characteristic curve of 0.78 (OR 95% CI: 0.77 to 0.83).[37]

The NAFLD score was calculated based on an algorithm including a logistic function with the presence of metabolic syndrome defined by criteria of International Diabetes Federation,[38] presence of type 2 diabetes, and fasting concentrations of insulin, aspartate aminotransferase (AST) and the AST/alanine transaminase (ALT) ratio:

NAFLD score=−2.89+1.18×metabolic syndrome (yes/no)+0.45×type 2 diabetes (yes/no)+0.15×fasting insulin (mU/L)+0.04×fasting AST (U/L)−0.94×AST/ALT.

Presence of hepatic steatosis was defined by a NAFLD score ≥−0.640, a value with a sensitivity of 86% and a specificity of 71%, when compared with proton MRI.[36]

## Assessment of covariates at baseline

Sociodemographic, lifestyle and health characteristics were collected by self-administered questionnaires. Age, sex, marital status, occupational status and educational level were included as indicators of sociodemographic condition. Smoking status was classified as 'never', 'former' and 'current'. Alcohol consumption was assessed by the number of alcoholic beverage units consumed in the past week and further categorised as 'abstainers' (0 unit/week), 'moderate' (1–21 units/week for men, 1–14 for women) and 'heavy' (>21 units/week for men, >14 for women) drinkers (1 unit corresponds to 8 g of alcohol). Physical activity was assessed with a self-administered quantitative physical activity frequency questionnaire.[39] Health characteristics included presence of metabolic syndrome and family history of diabetes. Anthropometric and blood pressure measurements were obtained using standard procedures and equipment as previously described.[29] Plasma triglycerides, high-density lipoprotein cholesterol and glucose were measured using standard enzymatic methods, and ALT, AST and GGT were measured using reference methods as standardised by the International Federation of Clinical Chemistry.

## Statistical analysis

Statistical analyses were performed using Stata (V.15; StataCorp, College Station, Texas, USA) with a two-sided test with α=0.05. Descriptive statistics were obtained in the participants included in this study in comparison with those excluded from this study. Cohen's kappa statistics were calculated to assess the agreement between the FLI and NAFLD score.

MDS, as a measure of adherence to the Mediterranean diet, was evaluated both categorically (quintiles) and continuously scaled as 1 SD unit. The association of MDS with the risk of hepatic steatosis was assessed using multivariable-adjusted Poisson regression models with robust SEs and estimating risk ratios (RRs) and 95% CIs. Models were adjusted for age, sex, marital status, occupational status, educational level, smoking status, energy intake, total energy expenditure and date of dietary assessment (to adjust for seasonality). We further adjusted for BMI and waist circumference as potential confounders or factors on the causal pathway to assess the possible impact of overall and central adiposity on the association of the Mediterranean diet and hepatic steatosis.

Additionally, we also adjusted for changes in BMI categories between baseline and follow-up; for alcohol consumption (units/week); for clinical variables of metabolic risk (blood pressure >130/85 mm Hg (yes/no), triglycerides >1.7 mmol/L (yes/no), high-density lipoprotein level

<1.29 mmol/L for men and <1.03 mmol/L for women (yes/no), and glucose level ≥5.6 mmol/L (yes/no))[38]; and for family history of diabetes and metabolic syndrome (only for FLI) to examine their influence on the association of interest.

Possible interactions between MDS and age, sex, BMI and alcohol consumption were tested using the Wald test. Several sensitivity analyses were conducted to examine the robustness of the observed findings. First, to assess the role of alcohol consumption (as alcohol is a risk factor for fatty liver accumulation), we excluded the alcohol component from the MDS, while adjusting for alcohol consumption as a covariate. We took the same approaches for the other MDS components to assess the impact of each component on the observed associations. Second, we conducted separate analyses after excluding participants with BMI ≥30 kg/m$^2$; implausible energy intake (<500 or >3500 kcal/day in women and <800 or >4000 kcal/day in men); excessive alcohol consumption; prevalent diabetes (defined as glycated haemoglobin ≥48 mmol/mol, or fasting plasma glucose ≥7.0 mmol/L, or use of hypoglycaemic drugs or insulin); or probable secondary causes of hepatic steatosis such as hepatitis B or C, HIV, hepatotoxic or autoimmune disease medications. We evaluated the robustness of the results to an alternative definition of prevalent hepatic steatosis. While we excluded participants with prevalent hepatic steatosis using the specified cut-offs of FLI or NAFLD score in the primary analysis, we used each of the two indices separately in sensitivity analyses, whereby we evaluated 2652 adults in longitudinal analysis based on FLI; and 2568 adults, based on NAFLD. Finally, we used more restrictive cut-points and excluded participants with NAFLD score ≥−0.640 or with FLI>30.

In a post-hoc analysis, due to inconsistency of the associations observed for FLI and NAFLD score, we also calculated the Hepatic Steatosis Index (HSI)[40] based on the ratio of AST/ALT, BMI, presence of type 2 diabetes and sex:

HSI=8×AST/ALT+BMI+2 (presence of diabetes)+2 (if women).

Presence of hepatic steatosis was defined by an HSI >36. After excluding participants with HSI >36 at baseline (n=2674), we evaluated 2351 adults.

In a post-hoc analysis pertaining to sensitivity of the results to model covariates and for better understanding of potential mechanisms, longitudinal associations of MDS with follow-up measures of log-transformed GGT, ALT and AST levels, and with changes in BMI and waist circumference from baseline to follow-up were examined using multivariable-adjusted linear regression. These results were expressed as β coefficient (95% CIs) for changes in each measure per 1 SD difference in MDS.

## RESULTS

### Participant characteristics

Of the initial 5064 participants, 2776 (54.8%) were excluded, leaving 2288 participants (65.4% women;

55.8±10.0 years) for analysis. Participants included were more likely to be women, show higher sociodemographic characteristics and lower BMI, waist circumference, liver enzymes or prevalence of metabolic syndrome in comparison with excluded individuals (online supplemental table S1). Being in the highest quintile of MDS was higher among women compared with men, positively correlated with sociodemographic characteristics, and negatively correlated with being current smokers, heavy alcohol drinkers, and with BMI, waist circumference, GGT and TG (table 1).

### Adherence to the Mediterranean diet and risk of hepatic steatosis

After a mean 5.3 (SD: 0.5) years of follow-up, there were 153 (6.7%) and 208 (9.1%) participants with hepatic steatosis based on FLI and NAFLD score, respectively (online supplemental table S2). Case identification by FLI and NAFLD score was modestly concordant (kappa=0.60).

Multivariable-adjusted analysis showed an inverse association between MDS quintiles and risk of hepatic steatosis based on FLI ($p_{trend}$ <0.006) with RR (95% CI) comparing the top to the bottom category of 0.50 (0.28 to 0.91). The inverse associations across quintiles of MDS weakened after adjustment for BMI ($p_{trend}$=0.031) or both BMI and waist circumference ($p_{trend}$=0.034) (table 2): RR (95% CI)=0.61 (0.34 to 1.09) and 0.60 (0.34 to 1.08), respectively. In analyses using MDS as a continuous variable, the inverse association with risk of hepatic steatosis based on FLI (0.84 (0.73 to 0.96) per 1 SD of MDS) remained unchanged but getting imprecise after adjustment for BMI (0.85 (0.73 to 0.99)) and adjustment for both BMI and waist circumference (0.85 (0.71 to 1.02)). In sensitivity analysis, the magnitude of the inverse associations changed a little after further adjustment for alcohol consumption, presence of metabolic syndrome, changes in BMI categories or clinical variables (medication use or prevalent diseases), while adjustment for BMI and clinical variables increased SEs (online supplemental table S3).

Conversely, there was no association between MDS and the risk of hepatic steatosis defined by NAFLD score criteria, with RRs (95% CIs) ranging from 0.93 (0.82 to 1.05) to 1.00 (0.86 to 1.17) over different regression models (table 2 and online supplemental table S3).

### Interaction and sensitivity analyses

No significant interactions were found between MDS and age, sex, BMI or alcohol consumption on risk of hepatic steatosis ($p_{interaction}$ >0.05; results not shown). The contribution of each component of the MDS on risk of hepatic steatosis was assessed by sequential subtraction of components from the score (figure 1). Excluding the components of the MDS did not substantially affect the inverse associations with hepatic steatosis based on FLI; the magnitude of the associations remained reasonably stable, but it became weaker (p>0.05) after excluding

fruits, cereals, dairy products, red or processed meat, or alcohol.

In sensitivity analyses, when excluding the alcohol component from the MDS but adjusting for alcohol consumption as a covariate, the inverse associations between MDS and risk of hepatic steatosis based on FLI became weaker (online supplemental table S4). The primary results were not different when excluding participants with BMI ≥30 kg/m$^2$, excessive alcohol consumption or secondary causes of hepatic steatosis (online supplemental table S4). Excluding participants with implausible energy intakes weakened the associations (online supplemental table S4). The analysis of an alternative definition of prevalent hepatic steatosis, excluding participants with only high FLI score at baseline, did not alter the significant inverse association between MDS and risk of FLI-based hepatic steatosis (online supplemental table S5). In post-hoc analyses, there was an inverse association between MDS quintiles and risk of hepatic steatosis based on HSI ($p_{trend}$=0.070) with RR (95% CI) comparing the top with the bottom category of 0.70 (0.55 to 0.91); a significant inverse association with HSI was observed per 1 SD increase in MDS (0.90 (0.82 to 0.98)) (online supplemental table S5). Effect sizes were of slightly higher magnitude when excluding those with FLI >30 or NAFLD score ≥−0.640 at baseline, but CIs were wider due to smaller sample size (online supplemental table S6).

For NAFLD score, no significant associations were found in any of the sensitivity analyses (online supplemental tables S4 and S5). The sole exception was when participants with a high NAFLD score at baseline were excluded, where an inverse association between MDS quintiles and risk of hepatic steatosis was present ($p_{trend}$=0.039), but this association was attenuated to the null after adjustment for BMI (online supplemental table S5).

### Longitudinal analyses for adiposity and markers of hepatic function

In post-hoc exploratory analyses, there were inverse associations of MDS with changes in BMI (β coefficient (95% CIs) per 1 SD higher MDS of −0.08 (−0.15 to −0.02)) and in waist circumference (−0.33 (−0.61 to −0.06)) (online supplemental table S7). For the markers of hepatic function, MDS showed a trend toward inverse association with GGT levels ($p_{trend}$=0.047) (β coefficient (95% CIs) per 1 SD higher MDS of −1.66 (−3.73 to 0.41)), but not with ALT or AST levels (online supplemental table S8).

### DISCUSSION

In this first population-based European study among adults free from clinically manifest hepatic steatosis to report on the prospective association between adherence to the Mediterranean diet and risk of hepatic steatosis, we found an inverse association between MDS and risk of hepatic steatosis based on FLI criteria. This relationship was attenuated to the null when controlled for general and central adiposity assessed by the BMI and

**Table 1** Baseline characteristics of participants according to quintiles of the Mediterranean diet score, CoLaus Study, Switzerland (n=2288)

| Characteristic | Quintiles of Mediterranean diet score* | | | | | P value* |
| | Q1 (n=458) | Q2 (n=458) | Q3 (n=457) | Q4 (n=458) | Q5 (n=457) | |
|---|---|---|---|---|---|---|
| Age (years) | 57.1±10.6 | 56.7±9.9 | 56.4±10.5 | 55±9.8 | 53.7±8.9 | <0.001 |
| Women (%) | 59.0 | 62.9 | 67.6 | 66.6 | 71.1 | 0.001 |
| Marital status (%) | | | | | | 0.51 |
| Single | 17.7 | 14.2 | 16.8 | 17.5 | 16.4 | |
| Married/cohabitant | 53.9 | 55.7 | 59.1 | 57.2 | 57.3 | |
| Widowed/separated/divorced | 28.4 | 30.1 | 24.1 | 25.3 | 26.3 | |
| Employed (%) | 57.9 | 60.3 | 59.1 | 68.3 | 72.0 | <0.001 |
| Education (%) | | | | | | |
| University | 19.2 | 21.6 | 26.0 | 31.7 | 31.1 | |
| High school | 26.4 | 28.2 | 28.4 | 29.7 | 29.5 | |
| Apprenticeship | 39.3 | 35.2 | 34.8 | 28.2 | 28.4 | |
| Mandatory education | 15.1 | 15.1 | 10.7 | 10.5 | 10.9 | <0.001 |
| Current smokers (%) | 24.7 | 21.8 | 22.3 | 16.2 | 15.5 | 0.014 |
| Alcohol intake (%)† | | | | | | |
| Abstainers | 21.8 | 24.0 | 25.8 | 24.7 | 21.2 | |
| Moderate | 63.8 | 66.2 | 69.4 | 70.7 | 76.6 | |
| Heavy | 14.4 | 9.8 | 4.8 | 4.6 | 2.2 | <0.001 |
| Total energy intake (kcal/day) | 1819±705 | 1812±675 | 1781±729 | 1821±653 | 1801±595 | 0.87 |
| Protein (% energy) | 15.8±3.3 | 15.9±4.1 | 15.3±3.0 | 15.2±3.0 | 14.7±2.6 | <0.001 |
| Carbohydrates (% energy) | 45.3±9.2 | 45.9±9.1 | 47.3±8.0 | 47.6±8.2 | 49.0±7.8 | <0.001 |
| Fat (% energy) | 33.6±6.5 | 34.5±6.9 | 34.6±6.7 | 34.5±6.5 | 34.1±6.9 | 0.11 |
| TEE (kcal/day) | 2562±589 | 2600±618 | 2564±602 | 2558±555 | 2589±565 | 0.84 |
| Metabolic syndrome (%)‡ | 11.8 | 9.8 | 12.3 | 12.7 | 7.4 | 0.063 |
| BMI (kg/m²) | 24.0±2.7 | 23.8±2.9 | 23.9±3.0 | 23.6±2.8 | 23.3±2.7 | 0.003 |
| Waist circumference (cm) | 85.4±8.5 | 84.9±9.4 | 85.0±9.2 | 84.0±8.7 | 82.9±8.6 | <0.001 |
| Triglycerides (mmol/L) | 1.1±0.5 | 1.0±0.5 | 1.1±0.5 | 1.0±0.5 | 1.0±0.5 | |
| Median (IQR) | 1.0 (0.8–1.3) | 0.9 (0.7–1.2) | 1.0 (0.7–1.3) | 0.9 (0.7–1.2) | 0.9 (0.7–1.2) | 0.009 |
| GGT (U/L) | 25.0±16.5 | 25.1±17.9 | 23.2±14.5 | 24.0±17.9 | 21.5±12.1 | 0.002 |
| Median (IQR) | 20 (15–29) | 21 (15–28) | 19 (14–28) | 19 (14–26) | 18 (14–25) | |
| ≥50 (%) | 7.4 | 6.3 | 5.7 | 6.1 | 2.6 | 0.025 |
| ALT (U/L) | 21.3±7.8 | 22.1±8.9 | 22.5±9.7 | 21.3±8.7 | 21.5±8.7 | 0.12 |
| Median (IQR) | 19 (16–25) | 20 (16–26) | 20 (16–27) | 19 (16–25) | 20 (16–24) | |
| ≥40 (%) | 3.5 | 5.4 | 8.3 | 4.6 | 3.5 | 0.005 |
| AST (U/L) | 26.0±5.85 | 26.2±6.5 | 26.1±6.6 | 25.8±6.1 | 25.7±5.7 | 0.68 |
| Median (IQR) | 25 (22–29) | 25 (22–29) | 25 (21–29) | 25 (22–29) | 25 (22–29) | |
| ≥37 (%) | 4.1 | 8.3 | 6.3 | 5.0 | 4.4 | 0.038 |

Data are mean ± SD for continuous variables or percent for categorical variables, unless otherwise stated.
*The population was divided into five groups by quintiles (Q1–Q5) of the Mediterranean diet score. P values were computed by using ANOVA for continuous variables and $X^2$ test for categorical variables.
†Alcohol consumption categorised as 'abstainers' (0 unit/week), 'moderate' (1–21 units/week for men, 1–14 for women) and 'heavy drinkers' (>21 units/week for men, >14 for women).
‡Metabolic syndrome defined according to the International Diabetes Federation (waist circumference ≥94 cm in men and ≥80 cm in women plus at least two of the following factors: serum triglycerides ≥1.70 mmol/L or specific treatment for this lipid abnormality; serum high-density lipoprotein cholesterol <1.03 mmol/L in men and <1.29 mmol/L in women or specific treatment for this lipid abnormality; systolic blood pressure ≥130 mm Hg or diastolic blood pressure ≥85 mm Hg or treatment for previously diagnosed hypertension; and fasting plasma glucose ≥5.6 mmol/L or previously diagnosed type 2 diabetes).
ALT, alanine aminotransferase; ANOVA, analysis of variance; AST, aspartate aminotransferase; BMI, body mass index; GGT, gamma-glutamyl transferase; IQR, interquartile range; TEE, total energy expenditure.

**Table 2** Prospective association of the Mediterranean diet score with the risk of hepatic steatosis, CoLaus Study, Switzerland (n=2288)

| | Risk ratio (95% CI) across quintiles of Mediterranean diet score* | | | | | P trend | Risk ratio (95% CI) per SD increase* |
|---|---|---|---|---|---|---|---|
| | Q1 | Q2 | Q3 | Q4 | Q5 | | |
| Range | 1.83–7.63 | 7.64–8.35 | 8.36–8.92 | 8.93–9.59 | 9.60–12.1 | | |
| N total | 458 | 458 | 457 | 458 | 457 | | |
| Fatty liver index, median (IQR)† | 21.8 (10.4 to 37.7) | 20.4 (9.0 to 39.4) | 18.5 (8.9 to 34.6) | 18.1 (7.7 to 33.1) | 14.2 (6.6 to 28.5) | | |
| N cases (score ≥60) | 36 | 43 | 35 | 22 | 17 | | |
| Unadjusted | 1.00 (ref) | 1.19 (0.78 to 1.82) | 0.97 (0.62 to 1.52) | 0.61 (0.37 to 1.02) | 0.47 (0.27 to 0.83) | 0.001 | 0.79 (0.70 to 0.90) |
| Multivariable‡ | 1.00 (ref) | 1.11 (0.72 to 1.72) | 1.07 (0.68 to 1.66) | 0.71 (0.42 to 1.22) | 0.50 (0.28 to 0.91) | 0.006 | 0.84 (0.73 to 0.96) |
| Multivariable+BMI | 1.00 (ref) | 1.12 (0.73 to 1.71) | 0.90 (0.58 to 1.39) | 0.74 (0.44 to 1.24) | 0.61 (0.34 to 1.09) | 0.031 | 0.85 (0.73 to 0.99) |
| Multivariable+BMI+WC | 1.00 (ref) | 1.07 (0.70 to 1.64) | 0.90 (0.59 to 1.37) | 0.74 (0.44 to 1.26) | 0.60 (0.34 to 1.08) | 0.034 | 0.85 (0.71 to 1.02) |
| NAFLD score, median (IQR)§ | −2.1 (−2.4 to −1.5) | −2.0 (−2.4 to −1.4) | −2.0 (−2.4 to −1.3) | −2.1 (−2.5 to −1.5) | −2.1 (−2.5 to −1.6) | | |
| N cases (score ≥−0.640) | 41 | 46 | 51 | 38 | 32 | | |
| Unadjusted | 1.00 (ref) | 1.12 (0.75 to 1.67) | 1.25 (0.84 to 1.84) | 0.93 (0.61 to 1.41) | 0.78 (0.50 to 1.22) | 0.17 | 0.93 (0.82 to 1.05) |
| Multivariable‡ | 1.00 (ref) | 1.13 (0.75 to 1.70) | 1.26 (0.85 to 1.87) | 1.01 (0.65 to 1.56) | 0.80 (0.50 to 1.28) | 0.28 | 0.95 (0.83 to 1.09) |
| Multivariable+BMI | 1.00 (ref) | 1.17 (0.78 to 1.75) | 1.17 (0.80 to 1.71) | 1.07 (0.69 to 1.65) | 0.95 (0.60 to 1.52) | 0.71 | 0.99 (0.86 to 1.15) |
| Multivariable+BMI+WC | 1.00 (ref) | 1.12 (0.75 to 1.67) | 1.17 (0.80 to 1.72) | 1.07 (0.70 to 1.65) | 0.96 (0.60 to 1.53) | 0.80 | 1.00 (0.86 to 1.17) |

Statistical analysis using Poisson regression with robust SEs; results are expressed as risk ratios and 95% CIs.

*In categorical analysis, the population was divided into five groups by quintiles (Q1–Q5) of the Mediterranean diet score; SD was 1.20 for the multivariable analyses.

†Calculated based on an algorithm including BMI, WC, triglycerides and gamma-glutamyl transferase.

‡Adjusted for age (years), sex, marital status (single, married/cohabiting and widowed/separated/divorced), occupational status (working and not working), education level (university, high school, apprenticeship and mandatory education), smoking status (never, former and current), energy intake (kcal/day), total energy expenditure (kcal/day) and date of dietary assessment.

§Calculated based on an algorithm including presence of the metabolic syndrome and type 2 diabetes, and concentrations of fasting serum insulin, fasting serum aspartate aminotransferase (AST) and the AST/alanine aminotransferase ratio.

BMI, body mass index; NAFLD, non-alcoholic fatty liver disease; WC, waist circumference.

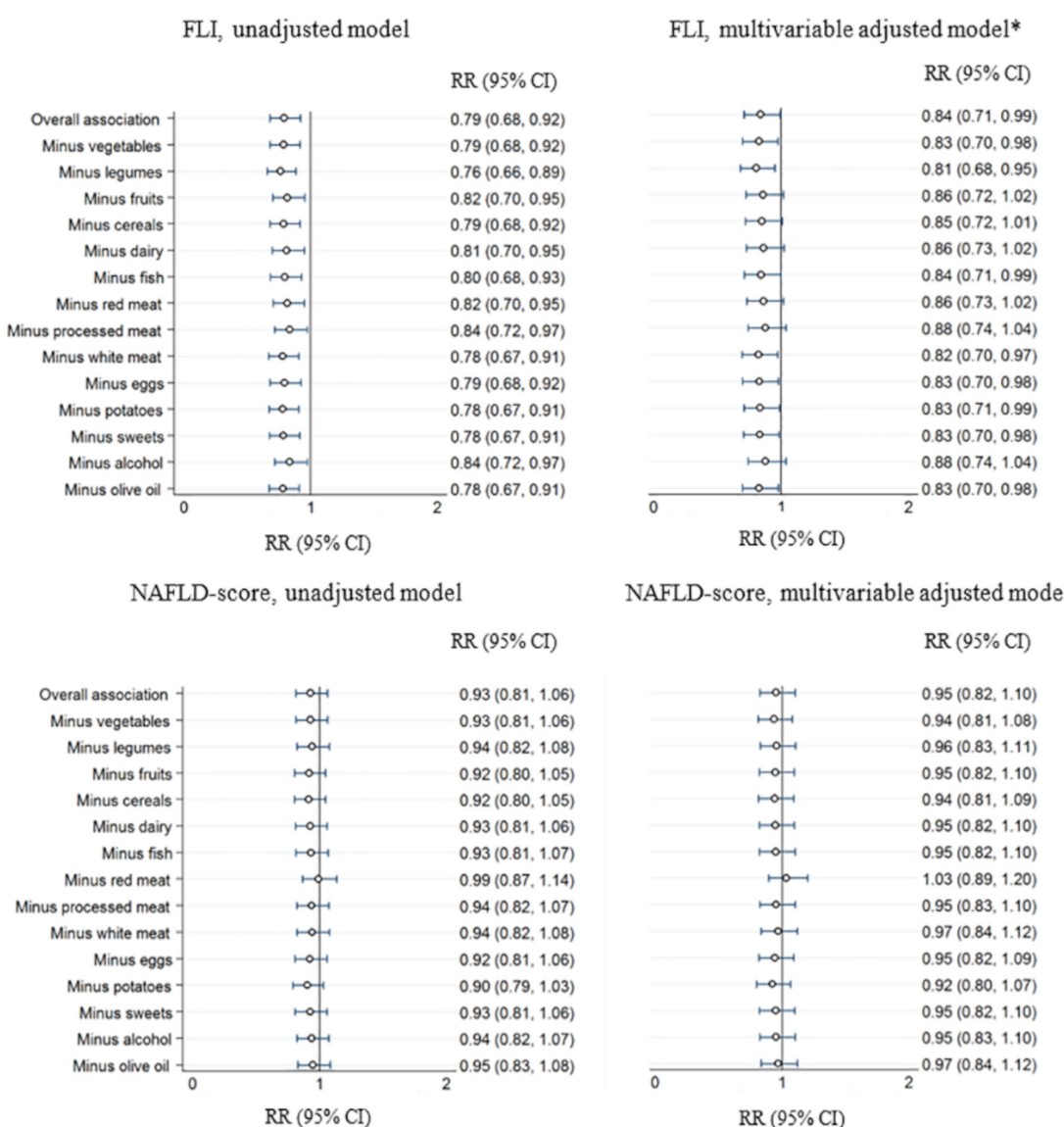

**Figure 1** Prospective association of the Mediterranean diet score (MDS) with the risk of hepatic steatosis, CoLaus Study, Switzerland (n=2288): sensitivity analysis to examine influence of each component of Mediterranean diet. Statistical analysis using Poisson regression with robust SEs; results are expressed as risk ratios (RRs) and 95% CIs. RRs and 95% CIs were estimated per 1 SD of MDS (overall association) or of each MDS computed after excluding one component. *Adjusted for age (years), sex, marital status (single, married/cohabiting and widowed/separated/divorced), occupational status (working and not working), education level (university, high school, apprenticeship and mandatory education), smoking status (never, former and current), energy intake (kcal/day), total energy expenditure (kcal/day) and date of dietary assessment. FLI, Fatty Liver Index; NAFLD, non-alcoholic fatty liver disease.

waist circumference. In contrast, there was no association between adherence to the Mediterranean diet and risk of hepatic steatosis based on NAFLD score criteria.

### Current findings in context of other evidence

Our finding based on FLI is consistent with the only published prospective study relating the Mediterranean diet to hepatic steatosis, where each SD increase in MDS was estimated to decrease the odds for incident hepatic steatosis by 26% (95% CI 10% to 39%).[28] By contrast, the point estimate of the effect size in our study was smaller (16% (95% CI 4% to 27%)). A possible explanation partly lies in methodological differences. We used biochemical and anthropometric markers to estimate hepatic steatosis

in the current study, while the previous study used CT assessment.

No association was found between adherence to the Mediterranean diet and risk of hepatic steatosis based on the NAFLD score. For possible explanations based on the differences in their components, the FLI includes GGT, while NAFLD score includes AST and AST/ALT ratio; the FLI includes adiposity markers, while the NAFLD score does not; the FLI includes a lipid marker (triglycerides) while NAFLD score includes markers of glycaemic status. Previous studies showed a modest association of GGT and ALT (but not AST) with the prevalence of hepatic steatosis.[1] Indeed, our analysis showed an inverse

association between MDS and GGT levels, but not with AST or ALT, and these findings could explain the discrepancy between the two indices. Notably, our sensitivity analyses using different definitions of hepatic steatosis showed an inverse association between MDS quintiles and risk of hepatic steatosis based on NAFLD score after excluding participants with high baseline NAFLD score only. This could be explained by modest concordance between the two measures. Of note, there were no statistically significant associations between MDS and risk of hepatic steatosis based on FLI after excluding participants with FLI >30 at baseline, which led to a smaller sample size and consequently a lower statistical power in our study. In a post-hoc analysis, we found an inverse association between adherence to the Mediterranean diet and hepatic steatosis as assessed by HSI, an alternative score to detect hepatic steatosis. Although case identification by HSI and FLI (kappa=0.27) or NAFLD score (kappa=0.18) was weakly concordant, our finding based on FLI is consistent with HSI. This could be explained potentially by BMI as one of the components of both FLI and HSI, highlighting the importance of obesity for incident hepatic steatosis.[41]

The inverse association between MDS and risk of hepatic steatosis defined by FLI remained significant after adjusting for BMI only, but became imprecise and not significant after adjusting for both BMI and waist circumference or for changes in BMI over the follow-up period. These results suggest the collinearity between the central adiposity and hepatic steatosis as the central adiposity may partly reflect hepatic steatosis. This biologically plausible finding is in agreement with previous cross-sectional findings for MDS and prevalent hepatic steatosis in CoLaus Study and the British Fenland Study we reported,[27] and a study in Hong Kong.[42] Our finding that the MDS was negatively associated with an increase in BMI after 5.3 years of follow-up is consistent with the findings from the Framingham Heart Study which observed the same trend over 6-year follow-up,[28] and from Nurses' Health Study and Health Professionals' Follow-up Study evaluating 20-year longitudinal data with repeated self-reported measures of diet and adiposity measures.[43]

Sequential subtraction of different components of the MDS showed that fruits, cereals, dairy products, red or processed meat, or alcohol partially accounted for the observed association. These findings agree with previous studies[26 28 44 45] including the Framingham Heart Study suggesting benefits of low consumption of red meat and high consumption of fruits or whole grains[28]; and a cross-sectional analysis from the PREDIMED Study suggesting the benefit of low consumption of red meat.[26] Moreover, the Mediterranean diet is characterised by a moderate-to-high consumption of whole grains, which has been inversely associated with the likelihood of having NAFLD.[46]

## Possible mechanisms and implications

Hepatic steatosis is associated with a number of metabolic risk factors including insulin resistance, type 2 diabetes, dyslipidaemia, metabolic syndrome and oxidative stress.[47] Mediterranean diet-associated phenolic compounds (phenolic acids and polyphenols) found in fruits and vegetables and high levels of monounsaturated fatty acids of olive oil have been shown to inhibit de novo lipogenesis, improve peripheral insulin sensitivity, and reduced cardiovascular risk mainly due to their antioxidant, anti-inflammatory and anti-fibrotic effects.[18 48–51] Moreover, different components of the Mediterranean diet, including omega-3 polyunsaturated fatty acids (PUFA), fibre and antioxidant rich-foods are inversely associated with hepatic steatosis.[52 53] One meta-analysis of interventional studies reported that omega-3 PUFA were negatively associated with hepatic steatosis.[54] The Mediterranean diet is also low in saturated fat, which has been demonstrated to increase hepatic triglycerides content and hepatic insulin resistance.[55 56] Finally, the high-fibre content of the Mediterranean diet has been associated with reduced hepatic fat.[18 52]

Hepatic steatosis is associated with cardiometabolic diseases and substantially impacts public health.[13 15 16] Thus, our finding of an inverse association between adherence to the Mediterranean diet and risk of hepatic steatosis would support the importance of dietary advice for the prevention of hepatic steatosis as well as its treatment. However, future work should confirm whether or not the clinical importance of the Mediterranean diet for the prevention of hepatic steatosis is independent of obesity or central adiposity.

## Strengths and limitations

To our knowledge, this is the first European prospective study assessing the association between the Mediterranean diet and risk of hepatic steatosis. The study had the benefit of a relatively large sample size and an average of 5.3 years of follow-up, and applied a definition of the Mediterranean diet that has been shown to be valid in a non-Mediterranean population.[33]

Several limitations of this study merit consideration. Measurement error and recall bias are inevitable when using self-reported dietary instruments, limiting the ability to precisely measure adherence to the Mediterranean diet, although adjustment for energy intake may have reduced the magnitude of measurement error.[57] We used diet data measured only at baseline but recognised that intraindividual variation over time might be present which would be expected to weaken the observed associations, and hence our findings may be biased towards the null. However, in CoLaus, average change in estimated total energy intake from first to second follow-up was 51 kcal/day and changes for each macronutrient (expressed as % of total energy intake) were about 1% (data not shown). Thus, dietary intake in CoLaus was relatively stable, suggesting that the lack of availability of repeat dietary measures is unlikely to alter our findings substantially.

Our ascertainment of hepatic steatosis was based on two indices, but not on liver biopsy or direct imaging assessment; hence, information bias (misdiagnosis) cannot be

ruled out. Although liver biopsy is the gold standard for diagnosing hepatic steatosis, its use in apparently healthy participants would be unethical and unfeasible in a large population-based study. Further, previous studies have shown that FLI and NAFLD score can accurately identify hepatic steatosis with good sensitivity and specificity, and these scores have been validated for use in large epidemiological studies.[36 37 58 59] But, although both FLI and NAFLD score can indicate hepatic steatosis, their precision is variable.[60] Moreover, presence of steatohepatitis or advanced liver fibrosis could influence the relationship of FLI or NAFLD score with hepatic steatosis.[61] Of note, our study was not designed to assess the role of adherence to the Mediterranean diet in hepatic fibrosis.

Although we adjusted for many relevant confounders and performed a series of sensitivity analyses, we cannot rule out residual confounding due to unmeasured variables or covariates measured with error. On the other hand, our adjustment for BMI and waist circumference as markers of general and central adiposity may potentially be an overadjustment if adiposity is on the causal pathway between dietary adherence and hepatic steatosis. However, since FLI may approximate hepatic steatosis with a degree of imprecision, adjusting for adiposity in these analyses may not represent adjusting the association between diet and steatosis directly, but through an HSI that already includes adiposity measures in its definition. Nevertheless, our analytical approach is comprehensive, showing the results for crude analyses, followed by multivariable adjustment without and with further adjustment for adiposity markers. Future research with repeat measurements should further investigate this issue. Generalisability of our findings is limited because included participants seemed to be healthier than those excluded, and our findings were obtained in a single European population. Still, they confirm the findings from a previous prospective study conducted in the USA[28] and might serve as a reference for other studies.

## CONCLUSION

Adherence to the Mediterranean diet was inversely associated with risk of hepatic steatosis based on the FLI, and the association was independent of several known risk factors. Conversely, the association was not observed when using different criteria specifying the NAFLD score. These findings support recommendations on following the Mediterranean diet for hepatic steatosis prevention in addition to the existing evidence for its benefit for CVD prevention. Nonetheless, the findings also highlight the need for further research with more accurate measures of hepatic steatosis to replicate these findings in different populations and settings.

**Author affiliations**
[1]Faculty of Medicine and Health, The University of Sydney School of Public Health, Sydney, New South Wales, Australia
[2]ARC Centre for Excellence in Population Ageing Research (CEPAR), University of Sydney, Sydney, New South Wales, Australia
[3]Department of Internal Medicine, Internal Medicine, University Hospital of Lausanne (CHUV), Lausanne, Switzerland
[4]Medical Research Council Epidemiology Unit, University of Cambridge, School of Clinical Medicine, Cambridge, UK

**Acknowledgements** The authors are grateful to all the participants and staff of CoLaus Study.

**Contributors** SK-S, FI and NGF designed the study question, and had full access to all the data in the study and took responsibility for the integrity and accuracy of the data. SK-S performed the statistical analyses and wrote the first draft with supervision from FI, PM-V and NGF. All authors contributed to interpretation of data, revised the article critically for important intellectual content and approved the final version of the manuscript.

**Funding** The CoLaus Study was and is supported by research grants from GlaxoSmithKline, the Faculty of Biology and Medicine of Lausanne, and the Swiss National Science Foundation (grants 33CSCO-122661, 33CS30-139468 and 33CS30-148401). SK-S was supported by the Swiss National Science Foundation (Doc.Mobility number P1LAP3-171805). NGF and FI acknowledge core MRC support (MC_UU_12015/5), and NGF acknowledges NIHR Biomedical Research Centre Cambridge: Nutrition, Diet and Lifestyle Research Theme (IS-BRC-1215-20014).

**Competing interests** None declared.

**Patient consent for publication** Not required.

**Ethics approval** The Institutional Ethics Committee of the University of Lausanne, which afterwards became the Ethics Commission of Canton Vaud (www.cer-vd.ch) approved the baseline CoLaus Study (reference 16/03); the approval was renewed for the first (reference 33/09) and the second (reference 26/14) follow-ups. The CoLaus Study was performed in agreement with the Helsinki Declaration and its former amendments, and all participants provided their written informed consent before entering the study.

**Provenance and peer review** Not commissioned; externally peer reviewed.

**Data availability statement** Data are available upon reasonable request. Non-identifiable individual-level data are available for researchers who seek to answer questions related to health and disease in the context of research projects who meet the criteria for data sharing by research committees. Please follow the instructions at https://www.colaus-psycolaus.ch/ for information on how to submit an application for gaining access to CoLaus data.

**ORCID iDs**
Saman Khalatbari-Soltani http://orcid.org/0000-0001-8437-1906
Pedro Marques-Vidal http://orcid.org/0000-0002-4548-8500
Fumiaki Imamura http://orcid.org/0000-0002-6841-8396
Nita G. Forouhi http://orcid.org/0000-0002-5041-248X

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
