## [Reviewer comments · BMJ Open]

ARTICLE DETAILS

TITLE (PROVISIONAL)	The association between adherence to the Mediterranean diet and hepatic steatosis: the Swiss CoLaus prospective cohort study
AUTHORS	Khalatbari-Soltani, Saman; Marques-Vidal, Pedro; Imamura, Fumiaki; Forouhi, Nita

VERSION 1 – REVIEW

REVIEWER	Ludovico Abenavoli University Magna Graecia of Catanzaro, Italy
REVIEW RETURNED	17-Jun-2020

GENERAL COMMENTS	- Introduction section: International guidelines agree to define a dietetic nutritional management to achieve weight loss, as an essential component of any therapeutic strategy. On the basis of its components, the literature reports the beneficial effects of the Mediterranean diet in preventing major chronic diseases, including obesity, diabetes, cardiovascular diseases, and some forms of cancers. In recent years, a growing body of evidences supports the idea that the Mediterranean diet, associated with physical activity and cognitive behavior therapy, may be the reference nutritional profile for the prevention and the treatment of NAFLD patients (PMID: 30033779; 32359737). I suggest also to well report the aims of this study - Discussion section: about the possible mechanisms of action, is important to underline that the patients with NAFLD have an increased prevalence of chronic diseases. Therefore, the treatment of NAFLD patients should be focused on reducing predisposing factors, such as insulin resistance, oxidative stress, and dyslipidemia. In this way, nutraceuticals have a pivotal role in the treatment of NAFLD. The adherence to Mediterranean diet characterized by the consumption of antioxidant-rich foods in general and of polyphenols in particular, is a part of the approach to treat NAFLD, and even a valuable instrument of prevention of this disorder (PMID: 29164049; 31003450)
--

REVIEWER	Nicolas Velasco Pontifical Catholic University Chile
REVIEW RETURNED	06-Jul-2020

GENERAL COMMENTS	The proposal from Kholatbari – Soltani relates to a very relevant item; the relationship between Mediterranean Diet (ascertained by a Mediterranean Diet Score = MDS), and the prevention of Hepatic Steatosis (HS). This study was made evaluating participants of the COLaus study, an ongoing population study, investigating several determinants of cardiovascular diseases at the city of Lausanne, Switzerland. The main result of the proposal is the inverse
---

	relationship between HS estimated by a Formula called Fatty Liver Index (FLI) and MDS. To estimate HS authors used two indirect formulas: FLI and NAFLD score. Authors recognize the use of those formulas as a limitation of their study, because the most proper methods to measure HS are liver biopsy or imaging methods. FLI or NAFLD score are widely used, but they could have a variable precision of its estimations (Zeiber - Sagis S. World J Gastroenterol 2013; 10: 57 – 64), and the presence of NASH or liver fibrosis could confound the relationship of FLI or NAFLD score with HS (Fedchuck L. Aliment Pharmacol Ther 2014; 40: 1209 – 22). The most important flaw of this paper is the time when they applied dietary records and calculated MDS. In methods section, authors state they applied dietary assessment only at the first follow up of their study, what is also the basal period of the study about relationship of FLI and NAFLD score with MDS. The prospective analysis to evaluate the former relationship were made after 5,3 years of follow up. There are no evidence to assure that the diet of the patients and its MDS, was the same as at 5,3 years before There are several tables were authors included adjust of data, and some of those results are very interesting. However, it is not appropriate to adjust FLI for any BMI, waist circumference (WC) or triglycerides, because all that items are also part of FLI. The same could be said to the compound called “clinical variables”, because among them is included triglycerides. The sum of tables, figures and supplemental tables and figures is twelve. The items shown in several tables are big. So, the big number of data shown make very hard to understand what authors want to highlight. The discussion section doesn't help to this aim. The reader may thank a more easy to read paper. In conclusion, is an interesting paper about and investigation item in which there is few research: the prevention of NAFLD by Mediterranean Diet. I think authors could overcome the defects shown in this comment.
--	--

REVIEWER	Federico Salomone Azienda Sanitaria Provinciale di Catania Catania, ITALY
REVIEW RETURNED	06-Jul-2020

GENERAL COMMENTS	This study by the Swiss CoLaus group has an important aim that is to demonstrate an inverse association between adherence to MedDiet and onset of fatty liver. Overall, the study is properly designed, methods are clear and statistical analysis seems rigorous. A relevant aspect of the study is the prospective design. I have the following criticisms: Maior  -FLI and the NAFLD fat score are not the only scores of liver fat. Since the two scores are discordant, I recommend to check also the hepatic steatosis index -The main predictor of mortality in NAFLD is fibrosis. For this reason authors should also check scores of fibrosis such as the NAFLD fibrosis score, the FIB-4 and APRI since these scores are predictors of mortality [quote PMID: 30111756] Minor
--

	-In the discussion section among the limits of the study authors should include the absence of robust measurement of fibrosis/steatosis such as Fibroscan/CAP -In the paragraph "Possible mechanisms and implications" authors should add the possible protective contribution of phenolic acids [see PMID: 32195455]
--	---

VERSION 1 – AUTHOR RESPONSE

Reviewer: 1

Reviewer Name
Ludovico Abenavoli

Institution and Country
University Magna Graecia of Catanzaro, Italy

Please state any competing interests or state 'None declared':
no

Please leave your comments for the authors below

1. **Introduction section: International guidelines agree to define a dietetic nutritional management to achieve weight loss, as an essential component of any therapeutic strategy. On the basis of its components, the literature reports the beneficial effects of the Mediterranean diet in preventing major chronic diseases, including obesity, diabetes, cardiovascular diseases, and some forms of cancers. In recent years, a growing body of evidences supports the idea that the Mediterranean diet, associated with physical activity and cognitive behavior therapy, may be the reference nutritional profile for the prevention and the treatment of NAFLD patients (PMID: 30033779; 32359737).**

Our response: Thank you for this comment. As suggested, we have now changed the relevant sentence in the Introduction section as follow:

Lines 12-15: *"The Mediterranean diet has been recently recommended for treatment of NAFLD [9]. In recent years, a growing body of evidence supports the idea that the Mediterranean diet may be the reference nutritional profile for the prevention of hepatic steatosis development [10–12]."*

We have also added the suggested references (#11 & 12).

2. **I suggest also to well report the aims of this study**

Our response: We have rewritten the aim of the study to make it more clear:

Lines 29-32: *"Given the limited evidence from population-based epidemiological studies thus far, we aimed to investigate the prospective association between adherence to the Mediterranean diet and the risk of developing hepatic steatosis among adults without clinically manifest hepatic steatosis."*

3. **Discussion section: about the possible mechanisms of action, is important to underline that the patients with NAFLD have an increased prevalence of chronic**

diseases. Therefore, the treatment of NAFLD patients should be focused on reducing predisposing factors, such as insulin resistance, oxidative stress, and dyslipidemia. In this way, nutraceuticals have a pivotal role in the treatment of NAFLD. The adherence to Mediterranean diet characterized by the consumption of antioxidant-rich foods in general and of polyphenols in particular, is a part of the approach to treat NAFLD, and even a valuable instrument of prevention of this disorder (PMID: 29164049; 31003450)

Our response: Taking on board your suggestions, we have now edited and improved the two paragraphs of the “possible mechanisms and implications” section as follows:

Lines 281-289: *“Hepatic steatosis is associated with a number of metabolic risk factors including insulin resistance, type 2 diabetes, dyslipidaemia, metabolic syndrome, and oxidative stress [46]. Mediterranean-diet associated phenolic compounds (phenolic acids and polyphenols) found in fruits and vegetables and high levels of monounsaturated fatty acids of olive oil have been shown to inhibit de novo lipogenesis, improve peripheral insulin sensitivity, and reduce cardiovascular risk mainly due to their antioxidant, anti-inflammatory, and anti-fibrotic effects [18,47–50]. Moreover, different components of the Mediterranean diet, including omega 3 polyunsaturated fatty acids (PUFA), fibre, and antioxidant rich-foods, are inversely associated with hepatic steatosis [51,52].”*

Lines 299-301: *“Thus, our finding of an inverse association between adherence to the Mediterranean diet and risk of hepatic steatosis would support the importance of dietary advice for the prevention of hepatic steatosis as well as its treatment.”*

We have also added the following three references to highlight your points:

47 Abenavoli L, Milic N, Di Renzo L, et al. Metabolic aspects of adult patients with nonalcoholic fatty liver disease. *World J Gastroenterol* 2016;22:7006–16. doi:10.3748/wjg.v22.i31.7006

49 Abenavoli L, Milic N, Luzzza F, et al. Polyphenols Treatment in Patients with Nonalcoholic Fatty Liver Disease. *J Transl Intern Med* 2017;5:144–7. doi:10.1515/jtim-2017-0027

50 Yang J, Fernández-Galilea M, Martínez-Fernández L, et al. Oxidative Stress and Non-Alcoholic Fatty Liver Disease: Effects of Omega-3 Fatty Acid Supplementation. *Nutrients* 2019;11. doi:10.3390/nu11040872

Reviewer: 2

Reviewer Name
Nicolas Velasco

Institution and Country
Pontifical Catholic University Chile

Please state any competing interests or state ‘None declared’:
None declared

Please leave your comments for the authors below:

The proposal from Kholatbari – Soltani relates to a very relevant item; the relationship between Mediterranean Diet (ascertained by a Mediterranean Diet Score = MDS), and the prevention of Hepatic Steatosis (HS). This study was made evaluating participants of the COLaus study, an ongoing population study, investigating several determinants of cardiovascular diseases at the city of Lausanne, Switzerland. The main result of the proposal is the inverse relationship between HS estimated by a Formula called Fatty Liver Index (FLI) and MDS.

- 1. To estimate HS authors used two indirect formulas: FLI and NAFLD score. Authors recognize the use of those formulas as a limitation of their study,**

because the most proper methods to measure HS are liver biopsy or imaging methods. FLI or NAFLD score are widely used, but they could have a variable precision of its estimations (Zeiber - Sagis S. *World J Gastroenterol* 2013; 10: 57 – 64 PubMed), and the presence of NASH or liver fibrosis could confound the relationship of FLI or NAFLD score with HS (Fedchuck L. *Aliment Pharmacol Ther* 2014; 40: 1209 – 22).

Our Response: We have added the following statements to the limitation section:

Lines 323-327: *“But, although both FLI and NAFLD-score can indicate hepatic steatosis, their precision is variable [62]. Moreover, presence of steatohepatitis or advanced liver fibrosis could influence the relationship of FLI or NAFLD-score with hepatic steatosis [63].”*

We have also added the suggested references (#62 & 63).

- 2. The most important flaw of this paper is the time when they applied dietary records and calculated MDS. In methods section, authors state they applied dietary assessment only at the first follow up of their study, what is also the basal period of the study about relationship of FLI and NAFLD score with MDS. The prospective analysis to evaluate the former relationship were made after 5,3 years of follow up. There are no evidence to assure that the diet of the patients and its MDS, was the same as at 5,3 years before.**

Our response: We agree that the using diet data only once at baseline is a potential limitation of this analysis. Despite the possible intra-individual variation in dietary habits over time, we cannot find a strong rationale to suspect that the variation would bias our results away from the null. In the CoLaus population, for example, undiagnosed hepatic steatosis is unlikely to have changed the dietary habit differentially and cause bias in a way to jeopardise our interpretation and conclusion. We have now added the following to the limitation section:

Lines 314-316: *“We used diet data measured only at baseline, and intra-individual variation over time might be present. However, we previously reported that dietary intake is stable in CoLaus study and in Switzerland in general [58,59].”*

We have also added the following two references:

58 Marques-Vidal P, Queteiro Fidalgo AS, Schneid Schuh D, et al. Lessons learned? Changes in dietary behavior after a coronary event. *Clin Nutr ESPEN* 2019;29:112–8. doi:<https://doi.org/10.1016/j.clnesp.2018.11.010>

59 Schneid Schuh D, Guessous I, Gaspoz J-M, et al. Twenty-four-year trends and determinants of change in compliance with Swiss dietary guidelines. *Eur J Clin Nutr* 2019;73:859–68. doi:10.1038/s41430-018-0273-0

- 3. There are several tables where authors included adjusted data, and some of those results are very interesting. However, it is not appropriate to adjust FLI for any BMI, waist circumference (WC) or triglycerides, because all those items are also part of FLI. The same could be said to the compound called “clinical variables”, because among them is included triglycerides.**

Our response: This is an important point with which we agree. In all tables, we show the results before and after adjustment. We are particularly interested in the causal pathway of the association of the Mediterranean diet with hepatic steatosis after adjustment for BMI and WC. Body mass and central adiposity are important risk factors for noncommunicable diseases. Clinicians and dietitians anyway focus on those factors regardless of the presence

of hepatic steatosis. Therefore, under the assumption that the adiposity is controlled (clinically or statistically), we would like to characterise and present the association of the MDS with the outcome. As would be predicted, the association was attenuated toward the null but remained significant. It is clinically important that adherence to the Mediterranean diet would link to hepatic steatosis regardless of the weight status.

- 4. The sum of tables, figures and supplemental tables and figures is twelve. The items shown in several tables are big. So, the big number of data shown make very hard to understand what authors want to highlight. The discussion section doesn't help to this aim. The reader may thank a more easy to read paper.**

Our response: We appreciate the suggestion to reduce down the quantity of our results presentations of tables and figures. We agree with the reviewer about the value of simplicity, but we also recognise potential limitations typical of observational research. Therefore, over and above typical observational studies, we wanted to thoroughly characterise the prospective association between adherence to the Mediterranean diet and hepatic steatosis and ensure the robustness of the observed findings. Thus, we would prefer to retain the current results, which include two tables and one figure in the main manuscript. We believe that the additional information included in the supplementary materials would be useful for those who are especially interested in greater detail but can be optional for those who choose not to access it. We hope this is acceptable as an approach to maintain the rigour we want to ensure.

We have now added a table of content on the first page of the supplementary file, which made the supplementary file more clearly marked for those who are interested in the extra information.

- 5. In conclusion, is an interesting paper about and investigation item in which there is few research: the prevention of NAFLD by Mediterranean Diet. I think authors could overcome the defects shown in this comment.**

Our response: Thank you. We trust that our responses to your helpful comments and our revisions to the manuscript now make it much improved.

Reviewer: 3

Reviewer Name
Federico Salomone

Institution and Country
Azienda Sanitaria Provinciale di Catania Catania, ITALY

Please state any competing interests or state 'None declared':
None

Please leave your comments for the authors below:

This study by the Swiss CoLaus group has an important aim that is to demonstrate an inverse association between adherence to MedDiet and onset of fatty liver. Overall, the study is properly designed, methods are clear and statistical analysis seems rigorous. A relevant aspect of the study is the prospective design. I have the following criticisms:

Major:

- 1. FLI and the NAFLD fat score are not the only scores of liver fat. Since the two scores are discordant, I recommend checking also the hepatic steatosis index.**

Our response: As suggested, we have performed a post-hoc analysis with hepatic steatosis measured based on hepatic steatosis index. We made the following changes:

Methods section (lines 146-151): *“In a post-hoc analysis, due to inconsistency of the associations observed for FLI and NAFLD-score, we also calculated the hepatic steatosis index (HSI) [40] based on the ratio of AST to ALT, BMI, presence of type 2 diabetes, and sex:*

$$HSI = 8 * AST/ALT + BMI + 2 \text{ (presence of diabetes)} + 2 \text{ (if women)}$$

Presence of hepatic steatosis was defined by a HSI>36. After excluding participants with HSI>36 at baseline (n=2674), we evaluated 2351 adults.”

Results are presented in table S5 and in text (Lines 204-208): *“In post-hoc analyses, there was an inverse association between MDS quintiles and risk of hepatic steatosis based on HSI ($p_{trend}=0.070$) with RR (95% CI) comparing the top to the bottom category of 0.70 (0.55, 0.91); a significant inverse association with HSI was observed per one SD increase in MDS [RR 0.90 (0.82, 0.98)] (Table S5).”*

We have inserted the following to the discussion section:

Lines 254-259: *“In a post-hoc analysis, we found an inverse association between adherence to the Mediterranean diet and hepatic steatosis as assessed by HSI, an alternative score to detect hepatic steatosis. Although, case identification by HSI and FLI ($kappa=0.27$) or NAFLD-score ($Kappa=0.18$) was weakly concordant; our finding based on FLI is consistent with HSI. This could be explained potentially by BMI as one of the components of both FLI and HSI, highlighting the importance of obesity for incident hepatic steatosis [41].*

- 2. The main predictor of mortality in NAFLD is fibrosis. For this reason, authors should also check scores of fibrosis such as the NAFLD fibrosis score, the FIB-4 and APRI since these scores are predictors of mortality [quote PMID: 30111756]**

Our response: This is an important point with which we agree, but our study was designed to examine the prospective association between adherence to the Mediterranean diet and future incidence of hepatic steatosis among those who did not have prevalent steatosis. Scores to assess fibrosis, such as the NAFLD fibrosis score, FIB-4 and APRI detect fibrosis and cirrhosis among patients with NAFLD and our study design did not allow assessment of participants with NAFLD for their health consequences. Therefore, we would like to keep our focus only on hepatic steatosis in this current manuscript. To explain this point, we have added the following statement to the limitation section:

Lines 326-327: *“Of note, our study was not designed to assess the role of adherence to the Mediterranean diet in hepatic fibrosis.”*

- 3. Minor -In the discussion section among the limits of the study authors should include the absence of robust measurement of fibrosis/steatosis such as Fibroscan/CAP**

Our response: We have added the following statement to the limitation section:

Lines 326-327: *“Of note, our study was not designed to assess the role of adherence to the Mediterranean diet in hepatic fibrosis.”*

4. In the paragraph "Possible mechanisms and implications" authors should add the possible protective contribution of phenolic acids [see PMID: 32195455]

Our response: Thank you. We have added the possible protective contribution of phenolic acid to the "possible mechanisms and implications" section. Please see our response to Reviewer #1, comment #3 where we included this point alongside other relevant text which reads as follows:

Lines 281-289: *"Hepatic steatosis is associated with a number of metabolic risk factors including insulin resistance, type 2 diabetes, dyslipidaemia, metabolic syndrome, and oxidative stress [46]. Mediterranean-diet associated phenolic compounds (phenolic acids and polyphenols) found in fruits and vegetables and high levels of monounsaturated fatty acids of olive oil have been shown to inhibit de novo lipogenesis, improve peripheral insulin sensitivity, and reduce cardiovascular risk mainly due to their antioxidant, anti-inflammatory, and anti-fibrotic effects [18,47–50]. Moreover, different components of the Mediterranean diet, including omega 3 polyunsaturated fatty acids (PUFA), fibre, and antioxidant rich-foods, are inversely associated with hepatic steatosis [51,52]."*

We have also added the suggested reference (#51).

VERSION 2 – REVIEW

REVIEWER	Nicolas Velasco Pontificia Univ Catolica Chile Chile
REVIEW RETURNED	24-Sep-2020

GENERAL COMMENTS	1. This study evaluated the relationship of Mediterranean Diet (MD) with hepatic steatosis (HS). The main flaw of the paper is the time they measured level of adherence to MD; only at the basal period. Author state: Lines 314-316: "We used diet data measured only at baseline, and intra-individual variation over time might be present. However, we previously reported that dietary intake is stable in CoLaus study and in Switzerland in general [58,59]." First reference (58), Clin Nutr ESPEN 2019;29:112-118, is a follow up of diet in Coronary Artery Disease patients (CAD), from 2009 to 2017. The evaluation was made in three groups: 1.History of CAD versus controls, 2. Patients studied before and after CAD and 3. Patients evaluated during incident CAD. There are several variations in the diet of CAD patients, and the conclusion of authors is: "In Switzerland, secondary prevention of CAD by diet is seldom implemented" Second reference (59), Eur J Clin Nutr 2019; 73:859-868, is a study to assess trends in compliance with dietary guidelines in the population of Geneva, Switzerland. 20310 patients were followed from 1993 to 2016 (patient's age 51,9 years, and 52,3% were women). During the period of study, there were some changes in the compliance of fruits, vegetables, meat, fresh fish and dairy products. Reading both papers, the reference 58 has no relationship with the in review HS paper. Reference 59 could have some relationship with in review HS paper, but this is only indirect and very light (CoLaus was made at Lausanne, not at Geneva, population from in review
---

	HS paper was older and with higher portion of women as compared to reference 59). It is not appropriate to state “However, we previously reported that dietary intake is stable in CoLaus study and in Switzerland in general” Also, authors of this study, associated with British investigator, evaluated the association of MD score with HS. CoLaus population included was from 2009 to 2013. The publication is very similar as the in review HS paper (BMC Medicine 2019;17:19). The MD score may be ascertained at the same time of HS evaluation. British investigators measured HS with ultrasound and FLI score. CoLaus, used FLI and NAFLD score. 2. Authors agree is not appropriate to adjust FLI with BMI, WC or Triglycerides, however they did it again. However they state: “We are in particular interested in the causal pathway of the association of the Mediterranean diet with hepatic steatosis after adjustment for BMI and WC. Body mass and central adiposity are important risk factors for non communicable diseases. Clinicians and dieticians anyway focus on those factors regardless of the presence of hepatic steatosis. Therefore, under the assumption that the adiposity is controlled (clinically or statistically), we would like to characterise and present the association of the MDS with the outcome. As would be predicted, the association was attenuated toward the null but remained significant. It is clinically important that adherence to the Mediterranean diet would link to hepatic steatosis regardless of the weight status” The rebuttal of authors is hard to understand. There is no reason to adjust a formula (as FLI is), using one of its components; as BMI, WC and triglycerides are. For better understanding, author should explain how they did adjust. It is possible to speculate adjust may be interesting to make if there were a evaluation of the correlation of FLI with other method to measure HS (ie. Ultrasound).
--	--

REVIEWER	Federico Salomone Azienda Sanitaria Provinciale di Catania, Catania, Italy
REVIEW RETURNED	23-Sep-2020
GENERAL COMMENTS	The authors responded adequately.

VERSION 2 – AUTHOR RESPONSE

Reviewer #3 (Federico Salomone)

Comment: The authors responded adequately.

Our Response: Thank you.

Reviewer #2 (Nicolas Velasco)

Comments

1. This study evaluated the relationship of Mediterranean Diet (MD) with hepatic steatosis (HS). The main flaw of the paper is the time they measured level of adherence to MD; only at the basal period. Author state:

Lines 314-316: “We used diet data measured only at baseline, and intra-individual variation over time might be present. However, we previously reported that dietary intake is stable in CoLaus study and in Switzerland in general [58,59].”

First reference (58), Clin Nutr ESPEN 2019;29:112-118 PubMed , is a follow up of diet in Coronary Artery Disease patients (CAD), from 2009 to 2017. The evaluation was made in three groups: 1. History of CAD versus controls, 2. Patients studied before and after CAD and 3. Patients evaluated during incident CAD.

There are several variations in the diet of CAD patients, and the conclusion of authors is: “In Switzerland, secondary prevention of CAD by diet is seldom implemented”

Second reference (59), Eur J Clin Nutr 2019; 73:859-868, is a study to assess trends in compliance with dietary guidelines in the population of Geneva, Switzerland. 20310 patients were followed from 1993 to 2016 (patient’s age 51,9 years, and 52,3% were women). During the period of study, there were some changes in the compliance of fruits, vegetables, meat, fresh fish and dairy products.

Reading both papers, the reference 58 has no relationship with the in-review HS paper. Reference 59 could have some relationship with in review HS paper, but this is only indirect and very light (CoLaus was made at Lausanne, not at Geneva, population from in review HS paper was older and with higher portion of women as compared to reference 59). It is not appropriate to state “However, we previously reported that dietary intake is stable in CoLaus study and in Switzerland in general”

Our Response: We thank the reviewer for their careful attention to the detail of the two cited papers (#58, #59), and we acknowledge the concerns raised in regard to direct applicability to the current CoLaus study. Therefore, we have now deleted references #58 and 59. We also deleted the sentence that we had previously added, but instead we have now added the following statements to address the point in question:

Page 19, lines 309-314 (Limitations section of the discussion):

“We used diet data measured only once at baseline but recognise that intra-individual variation over time might be present which would be expected to weaken the observed associations and hence our findings may be biased towards the null. However, in CoLaus, average change in estimated total energy intake from first to second follow-up was 51 kcal/day and changes for each macronutrient (expressed as % of total energy intake) were about 1% (data not shown). Thus, dietary intake in CoLaus was relatively stable, suggesting that the lack of availability of repeat dietary measures is unlikely to alter our findings substantially.”

2. Also, authors of this study, associated with British investigator, evaluated the association of MD score with HS. CoLaus population included was from 2009 to 2013. The publication is very similar as the in review HS paper (BMC Medicine 2019;17:19). PubMed The MD score may be ascertained at the same time of HS evaluation. British investigators measured HS with ultrasound and FLI score. CoLaus, used FLI and NAFLD score.

Our Response: We are pleased to be able to clarify this point. In the paper that has now been published in BMC Medicine, we used data from the Swiss CoLaus study and the UK-based Fenland study to investigate the association between the Mediterranean diet and hepatic steatosis prevalence, comparing two distinct populations. We cited this previous cross-sectional work in our current manuscript (ref

#27). As we highlighted in the Introduction section of our current under-review manuscript (page 6, lines 22-24), the cross-sectional design of these two studies limits inference for causal associations and can be used mainly for hypothesis generation. Thus, in this current study we aimed to investigate the prospective association between adherence to the Mediterranean diet and the risk of developing hepatic steatosis over follow-up time in a longitudinal analysis. To ascertain incidence, not prevalence, of hepatic steatosis, we excluded those who were considered as prevalent cases (FLI \geq 60 or NAFLD-score \geq 0.640 at baseline). This enabled us to examine the temporality of the association between diet at baseline and development of hepatic steatosis over follow-up time, in a standard epidemiological approach.

3. Authors agree is not appropriate to adjust FLI with BMI, WC or Triglycerides, however they did it again. However, they state:

“We are in particular interested in the causal pathway of the association of the Mediterranean diet with hepatic steatosis after adjustment for BMI and WC. Body mass and central adiposity are important risk factors for non-communicable diseases. Clinicians and dieticians anyway focus on those factors regardless of the presence of hepatic steatosis. Therefore, under the assumption that the adiposity is controlled (clinically or statistically), we would like to characterise and present the association of the MDS with the outcome. As would be predicted, the association was attenuated toward the null but remained significant. It is clinically important that adherence to the Mediterranean diet would link to hepatic steatosis regardless of the weight status”

The rebuttal of authors is hard to understand. There is no reason to adjust a formula (as FLI is), using one of its components; as BMI, WC and triglycerides are. For better understanding, author should explain how they did adjust. It is possible to speculate adjust may be interesting to make if there were an evaluation of the correlation of FLI with other method to measure HS (ie. Ultrasound).

Our Response: Thank you for the opportunity to clarify what we meant, and apologies that our prior response was unclear. In this current research we took the following analytical approach to examine the association of the Mediterranean diet pattern with incident hepatic steatosis.

Firstly, we conducted analysis in an unadjusted model for both indicators (FLI and NAFLD-score). This would give the interpretation of the crude association between the explanatory factor (diet) and the dependent variable (hepatic steatosis markers). Then, next, we conducted adjusted analysis in three different statistical models: (i) we accounted for a comprehensive range of potential confounding factors (summarised in methods section, and as footnote to Table 2); (ii) we additionally adjusted for BMI, and (iii) we further adjusted for both BMI and WC to account for both generalised and central obesity. We wanted to be entirely explicit so the readers can have information on all different levels from no adjustment through to different degrees of co-variate adjustment, and as such Table 2 includes all these results.

We focused on showing the effects on the risk ratio of adjustment for BMI or of BMI and WC because there is lack of clarity on whether adiposity may lie on the causal pathway (be a mediator) or be a confounder or neither for the association between diet and hepatic steatosis. Our results indicate that overall the effect is attenuated with greater adjustment. For instance, per SD change in diet score, for FLI the risk ratio increased from 0.79 unadjusted to 0.84 in the model adjusted for multiple potential confounding factors, but the attenuation was much more modest when additionally adjusting for BMI or both BMI and WC (risk ratio changed from 0.84 to 0.85). The interpretation is that accounting for adiposity had a minor impact over and above the multivariable adjusted model. This is indeed consistent with the argument put forward by the reviewer that since the outcome variable (e.g. FLI) already includes a marker of adiposity within it, therefore it is accounted for, but it is important to show this explicitly to the reader, and we have therefore shown the fuller results. The supporting argument for our approach is that because overall the index of hepatic steatosis is based on

several components, the specific contribution of the adiposity component is unclear as other component factors may exert bigger or smaller impacts at varying degrees of adiposity. Such an approach is also exemplified in analyses of other composite scores, such as the 'metabolic syndrome' or the Framingham riskscore. For instance, a previous study evaluated the Framingham Risk Score, calculated using multiple components including smoking status, but in the multivariable-adjusted model to examine the effect of physical activity on the risk score, the model included smoking status as a covariate (Patterson et al., BMJ Open 2020).

Reference:

Patterson F, Mitchell JA, Dominick G, et al Does meeting physical activity recommendations ameliorate association between television viewing with cardiovascular disease risk? A cross-sectional, population-based analysis BMJ Open 2020;10:e036507. doi: 10.1136/bmjopen-2019-036507

VERSION 3 – REVIEW

REVIEWER	Nicolas Velasco Pontificia Universidad Catolica de Chile, Chile
REVIEW RETURNED	30-Oct-2020

GENERAL COMMENTS	1. Author wrote a right sentence to overcome my observation about the time they measured Mediterranean diet score (only at basal period): “We used diet data measured only once at baseline but recognise that intra-individual variation over time might be present which would be expected to weaken the observed associations and hence our findings may be biased towards the null. However, in CoLaus, average change in estimated total energy intake from first to second follow-up was 51 kcal/day and changes for each macronutrient (expressed as % of total energy intake) were about 1% (data not shown). Thus, dietary intake in CoLaus was relatively stable, suggesting that the lack of availability of repeat dietary measures is unlikely to alter our findings substantially.” I think the previous limitation is important. Because that, it could be incorporated to the section “Strengths and Limitations” after the abstract. 2. Author consider correct to adjust by BMI and WC hepatic steatosis estimated by FLI . They state: “We focused on showing the effects on the risk ratio of adjustment for BMI or of BMI and WC because there is lack of clarity on whether adiposity may lie on the causal pathway (be a mediator) or be a confounder or neither for the association between diet and hepatic steatosis”. “The supporting argument for our approach is that because overall the index of hepatic steatosis is based on several components, the specific contribution of the adiposity component is unclear as other component factors may exert bigger or smaller impacts at varying degrees of adiposity”. In my opinion, the adjustment proposed is right when use a gold
---

	standard method to ascertain hepatic steatosis (MRI, TAC, ultrasound), and a formula, as FLI. In the paper authors didn't measure steatosis with a gold standard method, they estimate it by FLI then, if they adjust FLI by BMI or WC, they adjust FLI for one or two of its components, and not steatosis by BMI or WC.
--	--

VERSION 3 – AUTHOR RESPONSE

Reviewer #2 (Nicolas Velasco)

Comments

- 1. Author wrote a right sentence to overcome my observation about the time they measured Mediterranean diet score (only at basal period):**

“We used diet data measured only once at baseline but recognise that intra-individual variation over time might be present which would be expected to weaken the observed associations and hence our findings may be biased towards the null. However, in CoLaus, average change in estimated total energy intake from first to second follow-up was 51 kcal/day and changes for each macronutrient (expressed as % of total energy intake) were about 1% (data not shown). Thus, dietary intake in CoLaus was relatively stable, suggesting that the lack of availability of repeat dietary measures is unlikely to alter our findings substantially.”

I think the previous limitation is important. Because that, it could be incorporated to the section “Strengths and Limitations” after the abstract.

Our Response: Thank you for your comment. Due to the BMJ Open criteria, the Strengths and limitations section should contain only five bullet points, thus, to be able to include the limitation regarding diet data measurement we had to exclude one of the current points.

We decided to exclude the following bullet point (page 5; bullet point #5): “Generalisability is limited because our findings relate to a single European population.”

Please note that this limitation has been discussed in the limitation section of the manuscript (page 19, lines 329-330), so it is already covered.

We have now added the following bullet point instead (page 5; bullet points #4):

“We used dietary data measured only once at baseline, and intra-individual variation over time might be present which may weaken the observed associations towards the null; however, the dietary intake in CoLaus was relatively stable, suggesting that lack of repeated dietary measures is unlikely to alter our findings substantially.”

- 2. Author consider correct to adjust by BMI and WC hepatic steatosis estimated by FLI . They state:**

“We focused on showing the effects on the risk ratio of adjustment for BMI or of BMI and WC because there is lack of clarity on whether adiposity may lie on the causal

pathway (be a mediator) or be a confounder or neither for the association between diet and hepatic steatosis”.

“The supporting argument for our approach is that because overall the index of hepatic steatosis is based on several components, the specific contribution of the adiposity component is unclear as other component factors may exert bigger or smaller impacts at varying degrees of adiposity”.

In my opinion, the adjustment proposed is right when use a gold standard method to ascertain hepatic steatosis (MRI, TAC, ultrasound), and a formula, as FLI. In the paper authors didn't measure steatosis with a gold standard method, they estimate it by FLI then, if they adjust FLI by BMI or WC, they adjust FLI for one or two of its components, and not steatosis by BMI or WC.

Our Response: In our previous response to the original comment about this point, we had outlined our rationale for why we additionally adjusted for BMI and WC, including the citation of our detailed analytical approach as well as an example from the literature on an analogous approach for composite variables.

We think that our approach is comprehensive and lets the readers see the totality of the model building in a transparent way, from crude unadjusted analysis through to with adjustment for either BMI alone or both BMI and WC accounting for varying degrees of generalised and central obesity. Thus, we would prefer to retain the results from all statistical analysis models. This does explicitly show to the readers that accounting for adiposity had a minor impact over and above the multivariable adjusted model.

Since the reviewer is still concerned based on their point that we “adjust FLI for one or two of its components, and not steatosis by BMI or WC”, we have now inserted a statement in the discussion section (at lines 329-337 in the track changes version of the file), as shown here.

“On the other hand, our adjustment for BMI and WC as markers of general and central adiposity may potentially be an over-adjustment if adiposity is on the causal pathway between dietary adherence and hepatic steatosis. However, since FLI may approximate hepatic steatosis with a degree of imprecision, adjusting for adiposity in these analyses may not represent adjusting the association between diet and steatosis directly, but through a hepatic steatosis index that already includes adiposity measures in its definition. Nevertheless, our analytical approach is comprehensive, showing the results for crude analyses, followed by multivariable adjustment without and with further adjustment for adiposity markers. Future research with repeat measurements should further investigate this issue.”